# *Gardenia jasminoides* Extract, with a Melatonin-like Activity, Protects against Digital Stress and Reverses Signs of Aging

**DOI:** 10.3390/ijms24054948

**Published:** 2023-03-03

**Authors:** Morgane De Tollenaere, Emilie Chapuis, Jessy Martinez, Chantal Paulus, Joran Dupont, Eglantine Don Simoni, Patrick Robe, Bénédicte Sennelier-Portet, Daniel Auriol, Amandine Scandolera, Romain Reynaud

**Affiliations:** 1R&D Department, Givaudan France SAS, Route de Bazancourt, 51110 Pomacle, France; 2R&D Department, Naturex France Avignon (Givaudan), 250 Rue Pierre Bayle, 84140 Avignon, France; 3R&D Department, Givaudan France SAS, Bâtiment Canal Biotech 1, 3 Rue des Satellites, 31400 Toulouse, France

**Keywords:** digital stress, blue light, melatonin, *Gardenia jasminoides* fruit extract, skin, aging, crocin, crocetin–microbiota

## Abstract

Digital stress is a newly identified cosmetic stress that is mainly characterized by blue light exposure. The effects of this stress have become increasingly important with the emergence of personal digital devices, and its deleterious effects on the body are now well-known. Blue light has been observed to cause perturbation of the natural melatonin cycle and skin damage similar to that from UVA exposure, thus leading to premature aging. “A melatonin-like ingredient” was discovered in the extract of *Gardenia jasminoides*, which acts as a filter against blue light and as a melatonin-like ingredient to prevent and stop premature aging. The extract showed significant protective effects on the mitochondrial network of primary fibroblasts, a significant decrease of −86% in oxidized proteins on skin explants, and preservation of the natural melatonin cycle in the co-cultures of sensory neurons and keratinocytes. Upon analysis using in silico methods, only the crocetin form, released through skin microbiota activation, was found to act as a melatonin-like molecule by interacting with the MT1-receptor, thus confirming its melatonin-like properties. Finally, clinical studies revealed a significant decrease in wrinkle number of −21% in comparison to the placebo. The extract showed strong protection against blue light damage and the prevention of premature aging through its melatonin-like properties.

## 1. Introduction

Digital stress has been identified as a new source of stress by the cosmetic industry; it has become a subject of increasing interest along with the increase in daily exposure to screens. Moreover, several papers have demonstrated that screen exposure induces skin damage; however, the related mechanisms remain to be elucidated [1,2]. This stress is mainly represented by the blue light from light-emitting diodes in digital electronic devices such as smartphones, computer screens, tablets, and televisions. Indeed, the maximal absorption from these digital devices was at 450 nm, corresponding to the blue light range [3]. To be precise, blue light is characterized as high-energy visible light with a short wavelength within the range of 400 to 490 nm [4]. Blue light is close to the UVA spectrum and results in similar cutaneous disorders to those observed in photo-aged skin [5]. These disorders are now well described, and numerous studies have reported that blue light generates reactive oxygen species and thus causes oxidative damage, DNA damage, mitochondrial stress, hyperpigmentation in darker skin types, inflammation, and barrier disruption [1,2,3,4,5], all of which lead to premature aging.

Blue light is part of the visible light spectrum, and its negative effects on the body have been well documented [6]. Sleep disorders are the main negative consequence of exposure because visible light impairs the body’s internal clock. Various studies have demonstrated that visible light, especially blue light, has deleterious effects on the levels of melatonin, a crucial hormone secreted by the pineal gland and involved in day/night rhythms and thermoregulation [7,8]. Indeed, studies have reported a delay in, and partial suppression of, melatonin levels after long exposure to blue light (or visible light) [8,9,10,11,12].

Melatonin’s internal clock activity affects various tissues, including the skin. Some skin cells express a melatonin receptor (mainly MT1) and can synthesize melatonin, thus suggesting that skin has its own melatoninergic system [13,14,15,16,17]. Melatonin is an amphiphilic molecule with numerous beneficial effects on the skin. This pleiotropic molecule is a strong antioxidant that scavenges reactive oxygen species and stimulates the gene expression of enzymes involved in antioxidant defense. Through this antioxidant activity, melatonin decreases UV and DNA damage and protects the mitochondria [18]. Melatonin can stimulate wound healing, keratinocyte proliferation, and hair growth, and can inhibit melanogenesis and apoptosis [13,15]. Moreover, melatonin can slow cancer development. On the basis of these activities, melatonin is considered to be an anti-aging molecule [17].

Melatonin levels decrease with aging, and the amplitude of the difference between day and night levels decreases [19,20]. Therefore, blue light has a greater influence on younger than on older people [8]. Moreover, sleep deprivation is a factor affecting the skin and contributing to premature aging [21,22,23].

Thus, melatonin appears to be strongly correlated with blue light, and both play a role in skin aging.

The aim of this study was to evaluate the beneficial effects of *Gardenia jasminoides* fruit extract when stabilized in Natural Deep Eutectic Solvent (NaDES), which may act as a blue light filter and as a melatonin-like ingredient. Firstly, several experiments were conducted to prove the deleterious effects of digital stress on the skin and the protective effect of the plant extract. Secondly, an innovative co-culture model was designed to quantify and follow melatonin release and demonstrate its suppression by blue light.

In this work, we demonstrated that preserving the natural melatonin cycle is an innovative way to combat skin aging. Overall, these data explain how blue light affects skin aging in a process involving melatonin.

## 2. Results

### 2.1. Evidence of a Filtering Effect against Digital Stress

#### 2.1.1. A Physico-Chemical Barrier

Digital stress can be characterized mainly by exposure to blue light emissions between 400 and 490 nm.

The absorption spectrum of *Gardenia jasminoides* fruit extract in Figure 1 shows clear absorption in the blue light spectrum, with maximal absorption at 430 nm.

This result demonstrated that the fruit extract can act as a blue light filter.

#### 2.1.2. Protection of the Mitochondrial Network

The aim of this study was to demonstrate the beneficial effects of the filtering ability of the extract on a blue light-impaired mitochondrial network. After blue light irradiation, mimicking digital stress, the mitochondrial network was found to be significantly fragmented and diffuse (Figure 2). To obtain quantitative results, we segmented the mitochondrial networks via image analysis, which showed an organization involving trees and branches. The greater the number of trees and branches, the greater the degradation of the mitochondria network; the longer the network length, the lesser the fragmentation.

As shown in Table 1, after blue light-stress exposure, the network length, average tree length, and average branch length were significantly lower than those in untreated conditions (−43%, −68%, and −34%, respectively). In contrast, the number of trees and branches significantly increased (+180% and +46%, respectively). These results demonstrated that blue-light stress induced deleterious effects on mitochondria.

In the presence of the extract, the effects of blue light were significantly attenuated (Figure 2). A comparison of the active-treated samples with the blue light stress control indicated that the mitochondrial network length was nearly restored at the lowest dose (+51%) and was fully restored at 0.004% (+73%). The efficacy of the extract showed a dose response that significantly differed from that in the blue light stress control for both doses. In the presence of the extract, the number of trees and branches significantly decreased at both concentrations (down to −47% and −21%, respectively). Consequently, the average tree and branch lengths significantly increased (up to +94% and +31%, respectively). Therefore, we concluded that the extract acts as a shield against digital stress.

#### 2.1.3. Cell Spreading Analysis

Cell spreading, which is a good indicator of cellular stress, was analyzed using F-actin immunostaining to study the anti-blue-light protective effects of the extract. The model used in this experiment involved the culture of fibroblasts on a specific micropattern, facilitating cell-spreading observation. As shown in Figure 3, the representative images demonstrate that after blue light irradiation, the cell spreading decreased, showing a contracted F-actin cytoskeleton indicative of cell stress. In the presence of the extract, the broader spreading of cells and a relaxed F-actin cytoskeleton (Figure 3) were observed, thus indicating the protective effect of the extract against blue light.

To extend the analysis, we analyzed the stained cells in two ways. The cell area was measured and the percentage of cells with normal spreading was determined. Visually speaking, cells with correct spreading perfectly fit the micropattern (corresponding to an equilateral triangle).

As shown in Table 2, in comparison with untreated cells, blue light stress-treated cells showed significantly lower cell areas (−10%) and −26% well-spread cells. In the presence of the extract, cell areas were completely restored and even improved (+20%). The effect was significantly higher than that observed in the blue light stress control. The percentage of well-spread cells in the pre-treated samples significantly increased up to +57%.

The *Gardenia jasminoides* fruit extract efficiently protected the mitochondrial network by means of its blue-light-filtering ability.

#### 2.1.4. Skin Protection against Protein Oxidization

Digital stress-induced oxidative stress is comparable to that from UVA irradiation. We evaluated the ability of our *Gardenia jasminoides* fruit extract to protect against protein oxidation on skin explants after blue light exposure.

As shown in Figure 4, repeated exposure to blue light induced a significant increase in oxidized proteins (+93% **). In the presence of the extract, the level of detected oxidized proteins significantly decreased up to −86% **. The results confirmed the protective role of the *Gardenia jasminoides* fruit extract against blue light by means of its blue light-filtering property.

#### 2.1.5. Preservation of the Natural Melatonin Cycle

Blue light exposure delays the natural melatonin cycle. To fully investigate this important deleterious effect, we designed an innovative in vitro model combining a co-culture of keratinocytes and sensing neurons, with cyclization of the culture (day/night). Between D0 and D1, a second shock was induced to fully synchronize the cells and restart a new cell cycle. Cells were cultured under alternating day and night conditions, with exposure to blue light at the end of the day and night, to mimic chronic exposure to digital devices. Melatonin release was quantified for 3 days at several time intervals to observe its changes and the effects of chronic digital stress exposure.

As shown in Figure 5, in the untreated condition, melatonin release increased during the night phase on day 1. Cells exposed to blue light stress did not undergo induction of melatonin release. Finally, blue light stress significantly perturbed melatonin release in the co-culture model. These results demonstrated that our model was suitable for observing the melatonin cycle and subsequent blue light effects.

After demonstrating that blue light, mimicking digital stress, delays the natural cycle of melatonin release, we tested the effects of our active agent by using the same experimental conditions but while including pre-treatment with the extract. We focused our analysis on day 1, the window in which we observed a significant difference between the untreated condition samples and the blue light control condition samples. As shown in Figure 6, we compared these two conditions and the condition of those samples with those receiving extract treatment. In the presence of the extract, the release of melatonin was preserved at the same level as that in the untreated samples, whereas the blue light condition samples revealed impaired melatonin release. The difference between samples with the blue light control and the treatment with the extract was significant, thus indicating the efficiency of the extract in preserving the integrity of the natural melatonin cycle through its blue light-filtering property.

### 2.2. Microbiota Activation of Crocin, Forming a Melatonin-Like Molecule

We previously demonstrated that the extract protects the skin against digital stress by filtering blue light. Through this filtering effect, the melatonin cycle is preserved. This activity is driven by the presence of crocin in the extract, which absorbs blue light irradiation. Because crocin is a glycosylated form of crocetin, we hypothesized that the skin microbiota can convert crocin to crocetin.

#### 2.2.1. Activation of Crocin by the Skin Microbiota to Form Crocetin

A representative skin microbiota culture (sampled from swabs taken from several healthy donors) was incubated in the presence of crocin. The crocin conversion was visualized through analytical chemistry. As shown in Figure 7, crocin was metabolized and, after 66 h of culture, it began to increase. After 210 h, the crocin was fully converted to crocetin, the quantity of which appeared to have stabilized. The skin microbiota did not metabolize this final metabolite.

From our results, we confirmed that the crocin in the extract was converted by the skin microbiota into crocetin.

We then explored the potential efficacy of crocin, particularly crocetin, which was considered the active form of crocin. We hypothesized that crocetin acts as a melatonin-like molecule because its structure is similar to that of melatonin.

#### 2.2.2. Evaluation of the Affinity Score between Crocin Derivatives and the MT1 Receptor in Comparison to Melatonin via Molecular Docking Analysis

To investigate whether the extract acted as a melatonin-like molecule, we performed docking analysis on crocin derivatives, the main compounds in the extract (Figure 8). An affinity score was determined for the docking on the main skin receptor of melatonin, MT1, in comparison with melatonin.

In Table 3, the MT1 binding affinity score for melatonin was used as a reference to identify the baseline of the putative interaction with the MT1 receptor. We performed the same predictive analysis with crocetin and crocin. Only crocetin interacted with the MT1 receptor; the affinity scores were very close to those obtained with melatonin itself. However, the affinity score for alpha-crocin was negative, thus indicating poor interaction. The representative images in Figure 9 corroborated the results of the affinity scores: melatonin (a) fitted perfectly into the binding site of the receptor, and similar observations were made for crocetin (b). In contrast, the representative image of alpha-crocin (c) clearly showed that the molecule did not fit the binding site.

Crocetin appeared to be a good candidate as a melatonin-like molecule. Overall, these results clarified the mode of action of crocin, which is activated by the skin microbiota to form crocetin, which penetrates into the skin and acts as a melatonin-like molecule. These results revealed that *Gardenia jasminoides* fruit extract acts on the skin by means of two mechanisms: as a blue light filter that physically protects the skin against digital stress, and via activation by the skin microbiota after skin application, thus leading to the release of crocetin, a melatonin-like molecule that has anti-aging properties through MT1 receptor binding.

Melatonin is a powerful molecule with anti-aging properties. We then analyzed in vivo efficacy to confirm the observation that our extract acts as an anti-aging agent through its filtering effects and its similarity to melatonin.

### 2.3. Determination of Anti-Aging Efficacy through Quantification of Crow’s Feet Wrinkles in Volunteers Exposed to Chronic Digital Stress

Two groups of 20 volunteers applied a cream either containing or not containing the extract. The volunteers were exposed to digital stress for at least 4 h, at least 2 h of which occurred before bed. We quantified the number of hours of exposure to digital light for both groups to ensure that the blue light stress was equivalent. Then, the anti-aging properties of the product were examined by means of VISIA^®^ wrinkle analysis.

After 56 days of application, the number of wrinkles significantly decreased by −26% in the active group (Figure 10). In contrast, the group treated with the placebo did not show any significant decrease in wrinkles. A significant difference of −21% between groups was measured, thus confirming the anti-aging properties of the extract. Representative images are presented in Figure 11.

## 3. Discussion

Digital stress is not a cosmetic trend but is instead an external factor that has become increasingly present in human lives through increasing exposure to technology. Digital stress is mainly represented by exposure to blue light through radiation close to UVA that is emitted by the screens of electronic devices such as mobile phones, computers, or televisions [4] The consequences for the skin are well known, some of which are similar to those of UVA exposure. Numerous studies have related premature photo-aging to an increase in oxidative stress, DNA damage, and mitochondrial stress, among other conditions [1,2,5]. The aim of our work was to reveal the beneficial properties of the *Gardenia jasminoides* extract, regarding its ability to protect the skin against these direct deleterious effects of blue light irradiation. We selected a botanical extract that had specific absorption in the range of blue light (400 to 490 nm) and that could be used as a blue light filter to protect the skin against digital stress [24]. Moreover, this plant extract was traditionally used in folk medicine and in traditional Chinese medicine to treat inflammation, headache, edema, fever, hepatic disorders, and hypertension, evidencing its skin benefits [25,26]. Its anti-aging properties due to crocin content have been previously demonstrated but without there being a scientific link to digital stress and melatonin cycle preservation [27]. Rascalou and co-workers [3] have demonstrated that digital stress has dramatic effects on mitochondrial networks. Consequently, we evaluated the protective effects of our extract in an in vitro model of normal human dermal fibroblasts and observed the effects of blue light on the mitochondrial network. We demonstrated that the damage caused by blue light irradiation was significantly reversed in the presence of the extract. Indeed, the mitochondrial network was significantly less fragmented, and the cell spreading was significantly restored, thus indicating decreased cell stress. Therefore, the extract protected the mitochondria against digital stress.

Oxidation is another major consequence of digital stress; therefore, we evaluated the levels of oxidized proteins on skin explants after blue light irradiation. Oxidized proteins significantly increased after irradiation, as demonstrated by staining. In the presence of the extract when topically applied on skin explants, we observed a significant decrease by 81% and 86% with 0.002% and 0.004% extract, respectively. These results confirmed that the filtering effect of the extract provided strong protection against blue light.

Beyond the direct damage associated with irradiation, blue light perturbs the natural cycle of melatonin [8,9,10,11,12]. Melatonin is a powerful hormone involved in numerous biological functions and is a widely recognized anti-aging molecule [17]. Melatonin is also involved in day/night rhythms and the control of sleep [7,8]. Its perturbation, when triggered by blue light exposure, affects not only the overall body health and mental health but also the skin. Indeed, skin cells express melatonin receptors and, thus, can be affected by melatonin dysregulation. Because melatonin has numerous benefits when inside the cells [13,15,18], its natural cycle must be preserved. We designed an innovative co-culture model to visualize the effects of blue light on the melatonin cycle. As described in the literature [8,9], we observed that repeated exposure to blue light significantly depleted the level of melatonin, in comparison to the un-exposed condition. In the presence of the extract, the melatonin cycle was preserved, and its level was significantly higher than that in the blue light control.

Thus, we clearly demonstrated that the extract, through its filtering activity against blue light, protected the skin from oxidative damage and mitochondrial stress, and preserved the natural melatonin cycle.

In the second part of our investigation, we investigated the hypothesis that the extract might act as a melatonin-like molecule. Because skin microbiota have been widely purported to hydrolyze glycosylated compounds, we performed a study to verify this hypothesis [28]. Skin microbiota sampled from healthy donors were incubated in the presence of crocin. An analytical method was designed to follow the consumption of crocin and the appearance of crocetin, its deglycosylated form. According to the results, the skin microbiota was able to convert the crocin into crocetin. We then sought to identify the biological function of crocetin in the skin. To do this, we evaluated the potential docking of crocin and its deglycosylated form, crocetin, with the melatonin receptor, MT1 [16]. We demonstrated that crocetin had an affinity score close to that of melatonin, thereby indicating the potential melatonin-like activity of the extract.

We thus demonstrated that our extract protects against digital stress not only by acting as a filter protecting the skin against oxidative stress but also by preserving the natural melatonin cycle. We further demonstrated that this extract acts as a melatonin-like molecule after transformation by the skin microbiota. All effects led to anti-aging activity. We then extended our investigation to an in vivo study by designing an innovative experimental protocol in which volunteers were exposed for at least 4 h per day to digital devices (with 100% luminosity). After 56 days of a twice-daily application of the extract, we observed a significant decrease in the number of crow’s feet, whereas the placebo did not show any effect. The difference between the active and placebo groups was significant.

Thus, we conclude that the extract is a 2-in-1 cosmetic ingredient that can protect against digital stress, preserve the natural melatonin cycle, and act as a melatonin-like molecule that decreases the signs of aging.

## 4. Materials and Methods

### 4.1. Gardenia jasminoides Extract Description

The extract is a botanical cosmetic ingredient from crocin-rich Gardenia jasminoides J Ellis extract stabilized by a natural deep eutectic solvent (NaDES) composed of glycerin/betaine/water. Its composition is a mixture of crocin isomers.

#### 4.1.1. Extract Preparation

*Gardenia jasminoides* J Ellis (Rubiaceae) is cultivated in South China in Guangxi Province. For this study, the fruits were harvested in November. After drying and grinding, the plant was extracted in water (extract ratio 25–60:1). After filtration, to remove solids, the water extract was purified by column chromatography on a resin column with ethanol/water (70/30, *w/w*). The extract was then concentrated (to remove ethanol) and dried with maltodextrin as a carrier using a spray dryer to obtain a powder extract. The composition was *Gardenia jasminoides* extract/maltodextrin (95:5, *w/w*). The powder was titrated in crocin to 40% (*w/w*).

#### 4.1.2. Preparation of the NaDES Mixture

To produce 100 g of NaDES mixture, we mixed 40.4 g of glycerin, purchased from Cremer OLEO (UK) Ltd., Hamburg, Germany, 34.6 g of betaine purchased from Evonik (Tego Natural Betaine), and 25 g of water at 50 °C under stirring until homogenization and the complete dissolution of betaine.

#### 4.1.3. Final Preparation and Composition

To produce 100 g of the final product, 0.1 g of *Gardenia jasminoides* fruit extract (prepared as detailed above) was dissolved in 99.9 g of NaDES mixture (prepared as detailed above) at 50 °C until the complete dissolution of the powder extract; the mixture was then heated to 80 °C for 1 h.

The final extract is prepared with a specification of between 0.05 to 0.15% of the powder extract in a mixture of betaine, glycerin, and water.

Chromatogram and compounds identification are provided in Appendix A.

### 4.2. Mitochondrial Network and Cell Spreading Analyses after Digital Stress

Normal human dermal fibroblasts were incubated at 37 °C under 5% CO_2_ in CNTPRF medium (CELLnTEC, Bern, Switzerland) until reaching confluence. The cells were then treated for 24 h with two concentrations of *Gardenia jasminoides* fruit extract (0.002% and 0.004%, *v*/*v*) or were left untreated. Then the cells were loaded with MitoTracker Green dye (Thermo Fisher, Waltham, MA, USA) for 15 min. The cells were washed with PBS (Gibco, Thermo Fisher), detached, and seeded into a CYTOOplate™ with extra-large Y micropatterns at 2000 cells per well in 10% serum medium. After 1 h 30 min of incubation at 37 °C under 5% CO_2_ to allow the cells to spread, the medium was exchanged with a medium containing 1% serum and the extract at the two concentrations. After 2 h of treatment at 37 °C under 5% CO_2_, cells were irradiated with light-emitting diodes (reference: Kingbright KA-3529AQB25Z4S) at 447 nm for 1 h at 20 J/cm^2^, corresponding to a dose of 1 month (28 days) of digital screen exposure at 10 cm distance.

A Hoechst solution was added for 15 min in each well to stain the nuclei. The medium was replaced to wash off the Hoechst solution, then the cells were incubated in a medium with the extract. Live imaging was performed with a Leica microscope (Wetzlar, Germany) to observe the mitochondrial network on the basis of MitoTracker Green dye. At the end of the live imaging, the cells were fixed, and F-actin (Thermo Fisher) was stained with phalloidin 555 (Thermo Fisher). Images were acquired on the Operetta HCS platform (Perkin Elmer, Waltham, MA, USA).

The mitochondrial network was analyzed according to the sum of the lengths of all filaments of a single-cell network to generate the network total length, which was averaged across all single cells from the same well. The mitochondrial network was divided into groups of continuously linked filaments: this basal unit was called a tree. The number of trees per network, as well as their total lengths, were averaged across all single cells detected in each well. Each tree was divided into branches, defined at each end by either a junction or an endpoint. These branches were characterized on the basis of the measurement of their average and maximum lengths in the whole network of each single cell (average branch length). Image analysis was used to detect single cells on the micropattern and measure their area and spread. Correctly spread cells with an area exceeding 1800 µm² were counted.

### 4.3. Immunostaining of Oxidized Proteins on Cyclized Skin Explants after Digital Stress

Human skin explants prepared from an abdominal plasty coming from a 35-year-old Caucasian woman were stored in a survival BEM culture medium (BIO-EC’s Explant Medium) at 37 °C under 5% CO_2_. The skin explants were exposed every day to a dose of 63.75 J/cm² of blue light (λ_max_ = 455 nm), in 1 mL of HBSS medium, by using the Solarbox^®^ device to mimic digital stress for 4 days, resulting in 4 iterations of blue light exposure and a 255 J/cm² cumulative dose of blue light. The unirradiated skin explants were kept in 1 mL HBSS in the dark for the entire blue-light exposure time. At the end of the exposure, the medium was exchanged for BEM medium for all explants (2 mL). The extract diluted in water at 0.002% (*v/v*) and 0.004% (*v/v*) was topically applied every day before blue light exposure, and the medium was then renewed. After 4 days of incubation, immediately after the last blue light irradiation, then the skin explants were collected and frozen at −80 °C. The oxidized proteins were stained on frozen sections after pre-incubation with 2.4-dinitrophenylhydrazine (DNPH, Millipore, Burlington, MA, USA 90448) and incubation with anti-DNP antibody (Millipore 90451), diluted at 1:250 in PBS (Gibco, Thermo Fisher) and 0.3% (*w/v*) BSA (Sigma-Aldrich, Saint-Louis, MI, USA) for 1 h at 37 °C, with a biotin/streptavidin amplifying system (Thermo Fisher). The results were visualized with VIP, a violet substrate of peroxidase (Vector SK-4600, Vector Laboratories, Burlingame, CA, USA). The immunostaining was assessed via microscopic observation and pictures were taken using a Leica DMLB or Olympus BX43 microscope. Pictures were digitized with a numeric DP72 Olympus camera utilizing Cell ^D^ storing software.

Oxidized protein staining intensity was quantified with two open-source optical imaging software programs. Pictures (in .jpeg format) were opened using the GIMP-GNU image manipulation program. The strong- to light-pink color signals corresponding to the staining were selected, copied, and pasted into a new image and were saved as a .jpeg file, consisting solely of masks with the selected staining. This image was then opened in the ImageJ program. An area of the surface of interest was selected (in this instance, the dermis). Then, a histogram of the section was created that separated the total number of pixels in the image into 255 color categories spanning the visible spectrum. The peak corresponding to the strong- to light-pink color was determined by cutting and summing the appropriate counts from each picture. Then, the numbers corresponding to the peak were pasted into an Excel spreadsheet and summed.

The color index was then calculated by dividing the sum count by the surface area. The measurements were then expressed as a percentage of the blue light-stressed control condition.

All explants used in this study were obtained from surgical residues after written informed consent was obtained from the donors. The use of surgical waste did not require any prior authorization by an ethics committee.

### 4.4. Evaluation of Melatonin Release in a Cyclized Co-Culture of Sensory Neurons and Keratinocytes Exposed to Digital Stress

Sensory neurons derived from hiPS (human induced Pluripotent Stem cells) were seeded on six-well plates coated with Matrigel^®^ (Corning 354277, Boulogne-Billancourt, France, batch 72005017) at 250,000 cells per well with a differentiation medium. The differentiation medium consisted of DMEM-F12 (Panbiotech P04-41450, Aidenbach, Germany, batch 2730618) supplemented with 10% Knockout Serum Replacement (KSR)( Life Technologies, Thermo Fisher, 10828028, batch 1896527), 0.1 μM of retinoic acid (Sigma-Aldrich, St. Louis, MO, USA, R4643, batch SLBF3638V), a cocktail of central differentiation pathway inhibitors and 1% penicillin-streptomycin antibiotics (PS, Panbiotech P06-07100, batch 7631018). The cells were incubated for 6 days at 37 °C under 5% CO_2_, and the medium was changed every other day. The cells were then collected and seeded on 24-well plates coated with Matrigel^®^ at 100,000 cells per well in a differentiation medium. The cells were incubated for 3 more days at 37 °C under 5% CO_2_ in a differentiation medium. Then the medium was replaced by a maturation medium for sensory neurons, which was composed of DMEM-F12 supplemented with 1% N2 (Life Technologies 11520536, batch 2004543), BDNF at 10 ng/mL (PanBiotech CB-1115002, batch 051861), GDNF at 10 ng/mL (PeproTech, Thermo Fisher, 450-10, batch H170806), NT3 at 10 ng/mL (PeproTech 450-03, batch H171010), NGF at 10 ng/mL (Sigma N1408. batch SLBW7063) and 1% PS antibiotics. Cells were maintained in culture at 37 °C under 5% CO_2_. The culture medium was changed every other day. After 5 days of incubation, normal human epidermal keratinocytes were added to the plates above the differentiated hiPS cell layer and were seeded at 30,000 cells per well in culture medium consisting of 2:3 of medium for sensory neurons and 1:3 growth medium for keratinocytes (Lonza, Bâle, Switzerland, 192152, batch 723883). Cells were maintained in culture for 3 days at 37 °C under 5% CO_2_. The culture medium was changed every other day.

A cyclization protocol based on the use of glutamate and an increase in temperature (day phase) was then followed. In addition, two shocks with medium containing 50% FCS were performed for 2 h to synchronize the cell cycles [29,30,31,32,33]. The cultures were subsequently subjected for 8 h per day to conditions mimicking a day phase (glutamate 10 nM and temperature of 39.5 °C). The untreated culture with no cyclization was left in the dark at 37 °C. After 3 days of cyclization, the *Gardenia jasminoides* fruit extract at 0.004% (*v*/*v*) was added during the day phase. On the same day, 30 min before the night phase, the cells were exposed to blue light (λ = 450 nm; 20 mJ/cm²) to mimic exposure to digital stress. The extract was subsequently incubated again with each change in medium. After 24 h and 48 h, the culture supernatants were collected 30 min before the night phase, then 2 h, 5 h, and 8 h after the shift to the night phase, and were stored at −80 °C. At the end of the culture, the cells were washed once with PBS, and an MTT test was performed to validate cell viability.

ELISA was performed to determine the amount of melatonin released in the cell medium collected (ABIN511419, BlueGene, Shanghai, China).

### 4.5. In Vitro Study of Conversion of Crocin to Crocetin by the Skin Microbiota

#### 4.5.1. Microbiota Collection and Culture

Seven volunteers (three Caucasian women and four Caucasian men) showing no symptoms of skin disease were subjected to skin microbiota sampling on five body parts (forehead, cheek, nose, neck, and forearm). Ethics approval had been obtained and the study was approved by the Comité de Protection des Personnes (CPP), No. 2022-A02047-36, and No. SI 22.04050.000139.

Samples of skin microbiota were collected by non-invasive swabbing with sheets of 5 × 5 cm sterile gauze that had previously been soaked in 3 mL of 0.8% sodium chloride. All suspensions obtained by centrifugation of the impregnated gauze were pooled, filtered through a 40 µm filter, and centrifuged to remove large skin sample materials. The optical density at 600 nm (1 cm) was measured, and a value of 12.9 was obtained.

A 50 g·L^−1^ solution of *Gardenia jasminoides* fruit extract (batch 01A047-1EAA8869; Naturex, Avignon, France) was prepared in demineralized water, then filter-sterilized through a 0.22 µm filter. Its crocetin content (molar mass 976.97 g.mole^−1^) was 35% (molar content 358 mmoles.kg^−1^).

Flasks containing an appropriate culture medium (starch 10 g·L^−1^, yeast extract 1 g·L^−1^, meat extract 1 g·L^−1^ and bacto tryptone 2 g·L^−1^, pH 5.22) and 10% *Gardenia jasminoides* fruit extract (assay) or no extract (growth control) were inoculated with a volume of microbial suspension corresponding to 3% of the growth medium volume. A chemical stability control was also performed (with a non-inoculated mixture of culture medium and *Gardenia jasminoides* fruit extract).

The flasks were incubated for several days in a shaker incubator (30 °C, 115 rpm). The optical density was regularly measured at 600 nm. The carotenoid content was regularly determined via HPLC-UV analysis (cell removal by centrifugation; two-times dilution with 0.5 N sulfuric acid for stabilization and, finally, appropriate dilution with a 75% aqueous solution of DMSO).

#### 4.5.2. Stability and Crocin Conversion Study; HPLC Analysis

##### Chemical, Reagents, and Plant Materials

Trans-crocetin (purity 99.8% by HPLC; molar mass, 328.41 g·mole^−1^) was supplied by MedChemExpress (Sollentuna, Sweden). HPLC grade solvents and other reagents of analytical grade were from Merck KGaA.

##### Preparation of Standard Solutions

Trans-crocetin was solubilized in a mixture of 7.5 mL of DMSO and 2.5 mL of demineralized water, whereas the *Gardenia jasminoides* fruit extract solution was obtained with a DMSO, methanol, and water mixture (20/20/60, respectively).

##### Preparation of Sample Solutions

Acid-stabilized and conveniently diluted conversion samples were directly used for the HPLC-UV-MS analysis.

##### HPLC Instrumentation and Analysis

UltiMate 3000^®^ system from Thermo Scientific (Courtaboeuf, France) and an X-Bridge C18 column (150 mm × 4.6 mm; particle size, 5 µm; Waters, Guyancourt, France) with a standard guard cartridge for X-Bridge C18.

The elution gradient profile used for HPLC-UV and MS was as follows:Mobile phase: solvent A, ultrapure water; solvent B, methanol;Gradient: time, minutes/% A, 0/70, 20/20, 25/20, 26/70, and 30/70.

The injection volume was 10 µL. The column was equilibrated at 40 °C and the mobile phase flow rate was maintained at 1.0 mL.min^−1^.

Molecules were detected using an UltiMate 3000^®^ diode array detector with the wavelength set at 440 nm. Data were analyzed using Chromeleon software (Thermo Scientific; version 7.2).

##### Mass Spectrometric Conditions

An ISQ™EC Single Quadrupole Mass Spectrometer from Thermo Scientific was connected to the previously described UltiMate 3000^®^ system. Full-scan mass spectra were measured for mass-to-charge ratios (*m*/*z*) between 100 and 1000 in negative ion mode.

### 4.6. Molecular Docking Study of Crocin and Crocetin to the Melatonin Receptor MT1R

The crystallographic structures of crocin, crocetin, and melatonin were retrieved from the Protein Data Bank (www.rcsb.org, accessed on 21 November 2019). They were crystallized with four different agonists. The four models were derived for the docking studies after structure clean-up, consisting of the removal of solvent molecules and cations and the addition of hydrogen atoms, along with structure minimization. For each model, three sets of parameters for the docking (with increasing precision) were studied—Screen, Geom, and GeomX—corresponding to screening, accuracy, and exhaustive accuracy parameters. The software started with more conformations and a more accurate and detailed conformation search for Geom and GeomX. The best model was required to reproduce the experimental poses of the four reference agonists, and their docking score range was required to be consistent with their Ki ranges.

### 4.7. Analysis of Number of Crow’s Feet Wrinkles in Volunteers Exposed to Digital Stress

#### 4.7.1. INCI Formula

AQUA/WATER, CETYL ALCOHOL, GLYCERYL STEARATE, PEG-75 STERATE, CETEH-20, STEARETH-20, ISODECYL NEOPENTANOATE, ±GLYCERIN (and) BETAINE (and) WATER (and) GARDENIA JASMINOIDES FRUIT EXTRACT (and) MALTODEXTRIN, PHENOXYETHANOL, METHYL PARABEN, PROPYL PARABEN, ETHYL PARABEN, DIMETHICONE, FRAGRANCE, BENZYL SALICYLATE, LINALOOL, and D-LIMONENE.

#### 4.7.2. Study Design

A double-blind, inter-individual, and placebo-controlled clinical evaluation was performed on female volunteers, comprising two groups of 20 participants between 18 and 50 years of age. The volunteers met the inclusion and exclusion criteria, including having wrinkles on their faces and being in front of a screen (digital devices) for at least 4 h per day, including 2 consecutive hours during the evening at 100% of the digital devices’ luminosity. They were informed of the possible adverse effects of using the product and the technical conditions in which the assessment was performed. They willingly signed an informed consent form written in compliance with the Declaration of Helsinki and the act of 20 December 1988 of the Code de la Santé Publique.

One group of 20 volunteers tested the placebo cream, and a second group of 20 volunteers tested the cream containing 2% (*v*/*v*) of the extract. The treatment was applied twice daily for 56 days.

The anti-aging properties of the product were analyzed through the quantification of wrinkle numbers, using a VISIA^®^ 6th generation system. Digital photographs of the face were performed on days 0, 28, and 56. The control of the repositioning occurred directly on the data-processing screen, using an overlay visualization of the images at each time of acquisition. VISIA^®^ allows pictures to be taken with different types of illumination and very rapid image capture. A series of photos taken under multi-spectral imaging and analysis allowed for the capture of visual information affecting the appearance of the skin.

In this study, we analyzed crow’s feet wrinkles after 56 days of product application.

### 4.8. Statistical Analysis

All results are presented as mean ± standard error of the means of three independent triplicates. A Shapiro–Wilk test was performed to verify the normal distribution of data, following the Gaussian law. In a case where normality was passed, confirming parametric data, the mean values were compared with either an unpaired *t*-test (for two or fewer groups) or a one-way ANOVA, followed by a post hoc test (for more than two groups). In the case of non-parametric data, either a Kruskal–Wallis test, followed by a Mann–Whitney U test, was used for unpaired data, or a Wilcoxon test was used for non-parametric-paired data.

In all cases, results were considered significant, where *p* < 0.1 with #, *p* < 0.05 with *, *p* < 0.01 with ** and *p* < 0.001 with ***.

## Figures and Tables

**Figure 1 ijms-24-04948-f001:**
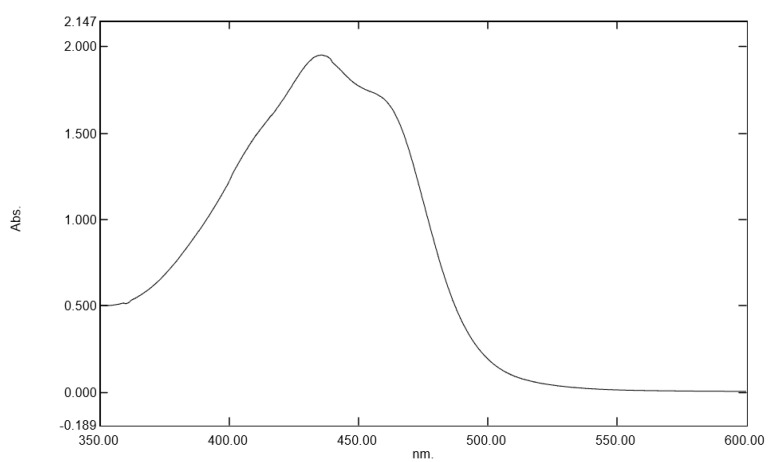
Absorption spectrum between 350 and 600 nm of *Gardenia jasminoides* fruit extract (dilution at 1/25 (*v/v*) in water).

**Figure 2 ijms-24-04948-f002:**
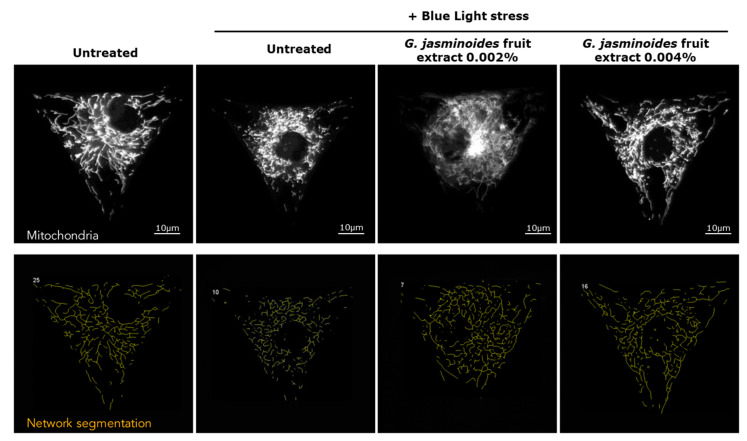
(**Top panel)** Representative images of mitochondrial network staining in normal human dermal fibroblasts that are either pre-treated or not pre-treated with the extract for 24 h. The blue light stress corresponds to irradiation at 447 nm for 1 h at 20 J/cm^2^. (**Bottom panel**) Network segmentation obtained from the staining. Magnification 40×.

**Figure 3 ijms-24-04948-f003:**
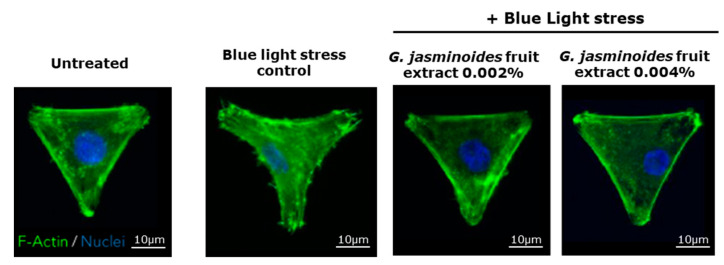
Normal human dermal fibroblasts that have been pre-treated or not pre-treated with the extract for 24 h. The blue light stress corresponds to irradiation at 447 nm for 1 h at 20 J/cm^2^. Representative images of F-actin immunofluorescence staining. Magnification 40×.

**Figure 4 ijms-24-04948-f004:**
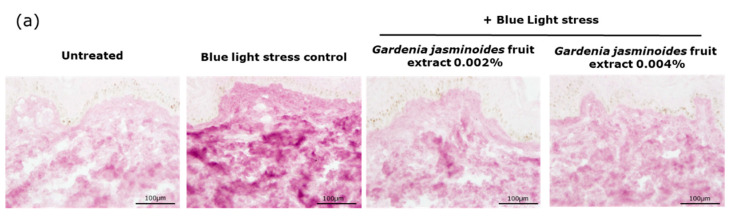
(**a**) Representative images of the oxidized protein staining of human skin explants that were treated or not treated with the extract and stressed or not stressed with 4 repeated blue light exposures at 63.75 J/cm². (**b**) Quantification of oxidized protein immunostaining. Statistical analysis with a Kruskal–Wallis ANOVA, followed by a Mann–Whitney U test in comparison to the blue light control, where * *p* < 0.05 and ** *p* < 0.01.

**Figure 5 ijms-24-04948-f005:**
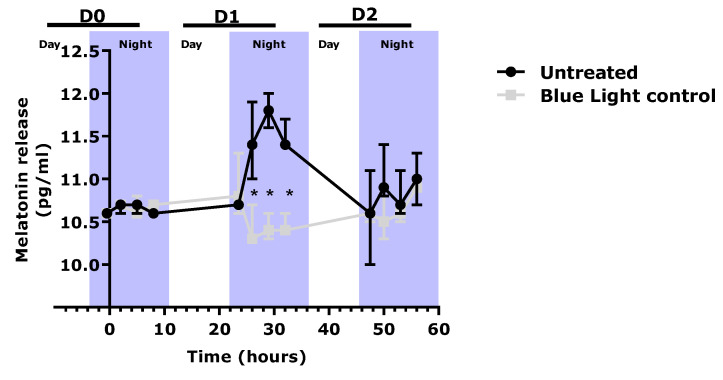
Quantification of melatonin release in the co-culture medium of cells exposed or not exposed to blue light stress (20 mJ/cm²). Statistical analysis with a Kruskal–Wallis ANOVA, followed by a Mann–Whitney U test in comparison to the blue light control, where * *p* < 0.05. Legend: D0 = Day 0; D1 = Day 1; D2 = day 2.

**Figure 6 ijms-24-04948-f006:**
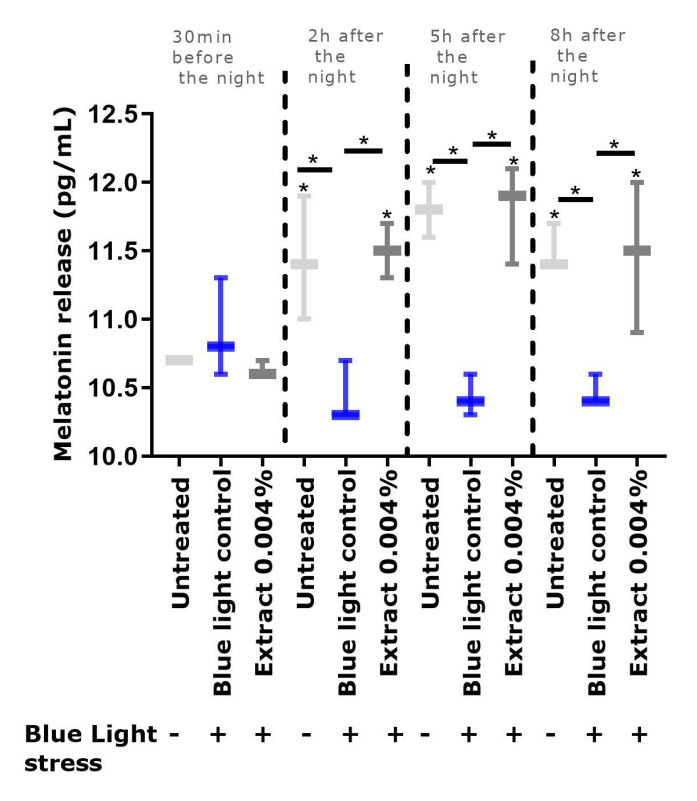
Quantification of melatonin release in the co-culture medium at day 1 in cells exposed or not exposed to blue light stress (20 mJ/cm²) and treated or not treated with the extract. Statistical analysis with a Kruskal–Wallis ANOVA, followed by a Mann–Whitney U test, in comparison with the blue light control, where * *p* < 0.05.

**Figure 7 ijms-24-04948-f007:**
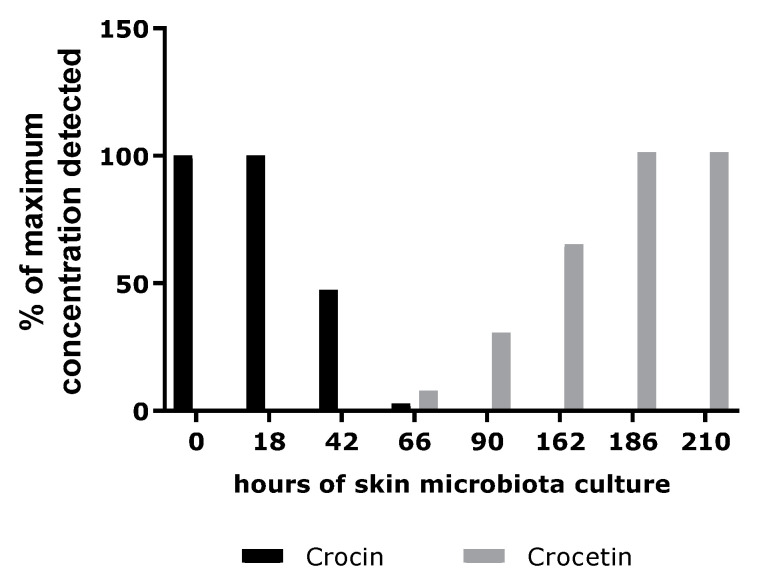
Dosage of crocin and crocetin in a skin microbiota culture with crocin.

**Figure 8 ijms-24-04948-f008:**
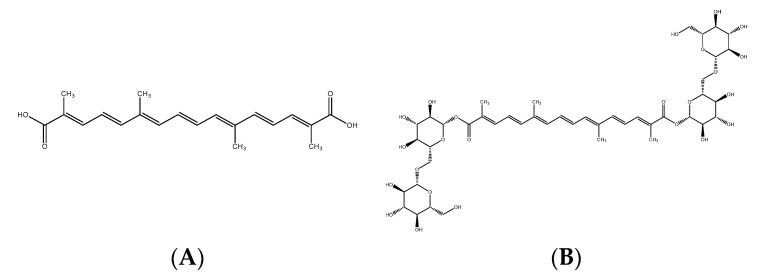
Structures of crocetin (**A**) and crocin (**B**).

**Figure 9 ijms-24-04948-f009:**
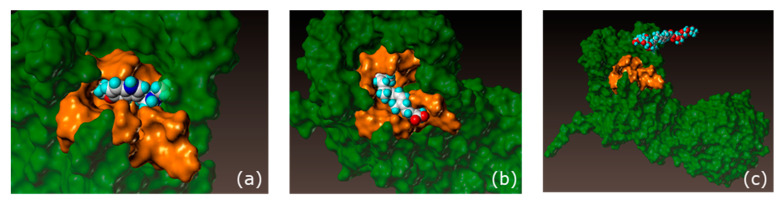
Representative images of melatonin (**a**), crocetin (**b**), and crocin (**c**) from the analysis of docking to the receptor MT1R. The green area represents the MT1 receptor surface, and the gold area represents the binding site of melatonin on the MT1 receptor. Carbon atoms are in white, oxygen atoms are in red, nitrogen in blue, halogens are in light blue (cyan).

**Figure 10 ijms-24-04948-f010:**
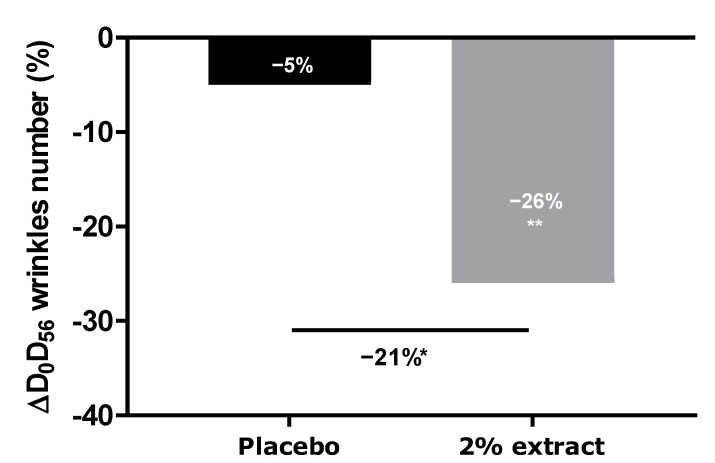
VISIA^®^ wrinkle analysis of the face after 56 days of application of a cream containing 2% or 0% (placebo) of the extract. A non-parametric paired Wilcoxon test was used to analyze the effects of the duration of application (D0 vs D56), and a non-parametric unpaired Mann–Whitney U test was used to compare products with a significant effect, with * *p* < 0.05 and ** *p* < 0.01.

**Figure 11 ijms-24-04948-f011:**
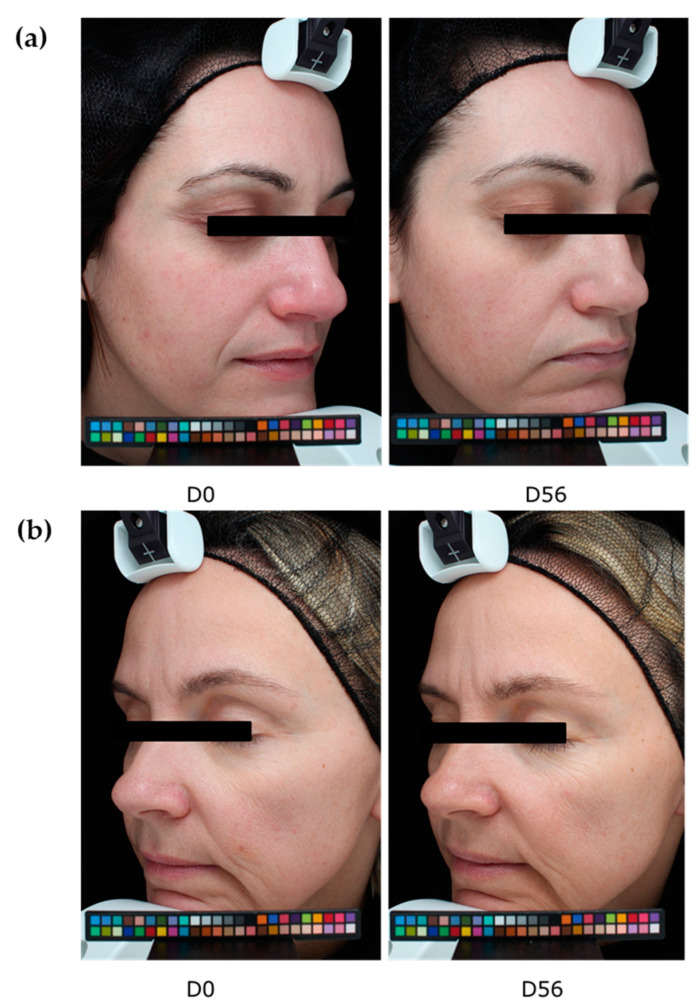
Representative images of volunteers who applied the extract at 2% (**a**) or the placebo (**b**) for 56 days. Colour palette is a calibration pattern.

**Table 1 ijms-24-04948-t001:** Network characterization and quantification. Significance was defined by a one-way ANOVA, followed by Dunnett’s multiple comparison test with a single pooled variance, where *** indicates *p* < 0.001.

Category	Untreated	Blue Light Stress Control	*Gardenia jasminoides* Fruit Extract at 0.002% + Blue Light Stress	*Gardenia jasminoides* Fruit Extract at 0.004% + Blue Light Stress
Network total length (µm)	454.06	259.69	391.89	450.41
		(−43% *** vs. untreated)	(+51% *** vs. blue light control)	(+73% *** vs. blue light control)
Number of trees/network total length	0.127	0.356	0.223	0.190
		(+180% *** vs. untreated)	(−37% *** vs. blue light control)	(−47% *** vs. blue light control)
Average tree length (µm)	8.79	2.79	4.48	5.40
		(−68%*** vs. untreated)	(+61% *** vs. blue light control)	(+94% *** vs. blue light control)
Number of branches/network total length	0.46	0.68	0.55	0.53
		(+46% *** vs. untreated)	(−19% *** vs. blue light control)	(−21% *** vs. blue light control)
Average branch length (µm)	2.02	1.33	1.72	1.75
		(−34% *** vs. untreated)	(+29% *** vs. blue light control)	(+31% *** vs. blue light control)

**Table 2 ijms-24-04948-t002:** Mean cell area and the percentage of normally spread cells. Significance was defined with a one-way ANOVA, followed by Dunnett’s multiple comparisons test with a single pooled variance, with *** *p* < 0.001.

Category	Untreated	Blue Light Stress Control	*Gardenia jasminoides* Fruit Extract at 0.002% + Blue Light Stress	*Gardenia jasminoides* Fruit Extract at 0.004% + Blue Light Stress
Cell area (µm²)	2088.71	1886.6	2257.17	2239.48
		(−10% *** vs. untreated)	(+20% *** vs. blue light control)	(+19% *** vs. blue light control)
Spread cells (%)	69.96	52.1	81.56	78.78
		−26% *** vs. untreated)	(+57% *** vs. blue light control)	(+51% *** vs. blue light control)

**Table 3 ijms-24-04948-t003:** Affinity scores.

Molecule	Affinity Score
Melatonin	5.6553
Crocetin	3.6232
Alpha-crocin	−36.4141

## Data Availability

Data are available on request because of restrictions, e.g., privacy or ethical issues.

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
