# Peer review of "Gardenia jasminoides* Extract, with a Melatonin-like Activity, Protects against Digital Stress and Reverses Signs of Aging"

_ijms, 2023, doi:10.3390/ijms24054948_

Round 1

Reviewer 1 Report

The authors present in this paper results of research on the effects of Gardenia jasminoides fruit extract/maltodextrin with 40 %w/w  crocin (considered to be the active ingredient) on human skin  irradiated by blue light ( LED 447 nm ,1h 20J/cm2 [ Fig 2,3], [Fig 4 ? nm, 63,75 J/cm2]. Blue Light as part of the natural solar spectrum (~400-525nm) is wellknown to create in the primary step of interaction with human skin free radicals (action spectrum) ! This paper demonstrates that the extract ( crocin) acts as an effective filter for the blue light wavelength used. The prevention of free radical formation is also confirmed by the stabilization of of the mitochondrial network, where obviously the nessecary amount of ROS, requiered for cell-wide mitochondrial oscillation have been reduced below threshold. Valuable also the observation of the transformation of crocin into crocetin, acting as melatonin like ingredient under skin miccrobiota conditions. This protective effect on natural melatonin cycle is a very interesting observation ! The results described and the methods used are worth to be published !

The authors however should rework the abstract and introduction ! Abstract Line 11, Line 29 "Blue light is a newly identified stress resulting from digital stress" BL is not newly identified as stressor! How is digital stress defined ? BL one possible component of digital stress ? Which exposure dose of BL are dangerous for skin / eye ?  You used comparable high doses 20 J/cm2, 63.75 J/cm2 . Is this exposure relevant to the average digital equipment user ? Line 15 : According to your paper you have not designed intentionally a melatonin like ingredient -you showed its existence ?! At the  moment you have no real evidence for the prevention of premature ageing through its melatonin like properties. 

Line 65 : " The aim of this study was to evaluate the deleterious ............and in vivo ." This was the aim ?????? Results ? Line 66 : Obviously the aim of this however was " to evaluate the beneficial effect ........Gardenia jasmonoides ..." Please rework carefully and  use the " digital stress and BL " only in a clear defined manner . Finally: The titel of the publication is not reflecting their content  and should be changed . There is per se no melatonin like ingredient in the extract [line 203] .Crocin is named to be the active ingredient ! There is evidence that the hormon melatonin like crocetin [line 320 melatonin cycle] protects against digital stress ?  Digital Stress [line 11,29] - BL ? Crocetin reverses the signs of ageing ?!

Reviewer 2 Report

The study:” A melatonin like ingredient protects against digital stress and reverses signs of ageing” is a well-designed study about the natural products that can be used against akin ageing. Authors described the effects of blue light on skin cells. Solar radiation, including recently recognized blue light, can have deleterious effects on living organisms exposed to sun, especially skin. Therefore, testing the direct effects of blue light can be very beneficial. The authors demonstrated marked melatonin-like protective effects of natural extract in skin cells exposed to blue light under laboratory conditions.

Few things are however unclear: How many times have you repeated the experiments? In fig 4 you described the reaped exposure to blue light. Please be more specific: time period of the exposure, cumulative dose, etc…

In fig 5 melatonin levels are low in untreated cells during nights of D0 and D2, yet high at D1. Why? How do you explain this phenomenon?

Also, how many replicas of the experiment presented in Fig 6.

Reviewer 3 Report

This manuscript has some merit -- the authors have performed a good variety of assays and tests to examine the merit of their plant extract for skin protection.

However, this paper is written like an advertisement, not like a scientific paper.  The Abstract is vague and subjective.  The plant material under investigation, Gardenia jasminoides, is not even mentioned in the abstract. An 'active ingredient' is mentioned here, and throughout the manuscript, but the reader much later discovers that what the authors are talking about is an extract from Gardenia jasminoides.  An extract is a complex mixture containing, usually, multiple phytoactives and other plant-derived compounds.   It probably contains active ingredient(s), but an extract is NOT an AI.  The abtract is also extremely vague -- not quantitative.  'a significant decrease in wrinkle #" how much?  quantify here, this is where the results must be established and summarized.  'strong protection'  what does that mean?  that is a subjective phrase that has no place in a serious scientific paper - quantify

Introduction.  Again, the 'AI' is vaguely mentioned but this is not an AI, it is a mixture, a plant extract.  Later they call it a 'molecule' (lines 67etc).  Again, an extract will be a mixture of compounds and mixture of molecules, but it is not a molecule.  Why are the constituents of G. jasminoides never described?  this much is already in the literature.

Results.  Fig. 1 there ought to be quantitative info in the figure legend.  'the greater the degradation' is vague.  Ditto for Table 1 'were significantly lower' does not cut it for a science journal.  Quantify. 

Why is there not even an HPLC of the plant extract given?  no analysis of the composition of the plant extract.  this is a basic minimum. Assume that crocin/crocetin are in there, but what concentration, what level, what else??  are there potentiating interactions with other compounds in the plant extract? 

M&M  What was the rationale for this company to examine this particular plant extract?  is there traditional knowledge, history of human use  or Traditional Chinese Medicine that supports the general hypothesis?

Figure 5.  Do D1 and D2 need to be identified in the legend.

small things.  The authors, once they finally DO identify the plant material, abbreviate it as Gardenia J.  This is botanically incorrect.  The correct abbreviation for the latin genus species should be G. jasminoides.

non science jargon.  We don't say 'perfectly fit' (line 128) - that is opinion.  Quantify.  "Cell areas were completely restored and even improved"  again, quantify.  significantly lower cell areas - not enough

Reviewer 4 Report

The study performed by De Tollenaere et al. is well performed, timely and interesting and merits publication. I only have a few comments to make and one major question.

Major question - On line 191 the authors report that it is crocin responsible in the extract that absorbs blue light....what proof is there of this?...a reference is required. The authors perhaps should have done some shielding experiments to prove that the extract actually absorbs blue light and that is it not the NaDES mixture in which it is incorporated as a whole that absorbs the blue light. There is no control in the experiments of the mixture with and without the extract to prove that it is the actual crocin-rich extract responsible for the effects observed. This should be discerned or clarified.

- The authors mention that 20 mJ/cm2 blue light corresponds to 1 month of blue light exposure, but this is quite meaningless if it is not reported how many h/day for a month. It could be 24 h a day for a month or 4 h/day for a month. So this needs to be defined better.

- Line 363: the authors mention the 10% medium was exchanged with less serum. How much less?? A % should be given for this serum.

- Line 386: authors mention skin explants...but there is no mention of where these explants come from/purchased and what kind.......human???

- Why were two different blue-light sources used for the dermal fibroblast experiments and for the skin explant experiments?

Round 2

Reviewer 3 Report

please pay attention to abstract wording -- you did not "design" an ingredient, you extracted it from a plant source

Firstly, 74 several experiments were designed to prove the deleterious effects of digital stress in skin 75 and the protective effect of the plant extract. Secondly, an An innovative co-culture model 76 was also designed to quantify and follow melatonin release, and demonstrate its suppres- 77 sion by blue light.   not a proper word for an English speaker

The results are better demonstrated and justified in this revision.
